# Iodine Bioavailability and Biochemical Effects of *Brassica oleracea* var. *sabellica* L. Biofortified with 8-Hydroxy-7-iodo-5-quinolinesulfonic Acid in Wistar Rats

**DOI:** 10.3390/nu16213578

**Published:** 2024-10-22

**Authors:** Joanna Krzemińska, Ewa Piątkowska, Aneta Kopeć, Sylwester Smoleń, Teresa Leszczyńska, Aneta Koronowicz

**Affiliations:** 1Department of Human Nutrition and Dietetics, Faculty of Food Technology, University of Agriculture in Krakow, al. Mickiewicza 21, 31-120 Krakow, Poland; joanna.krzeminska@urk.edu.pl (J.K.); ewa.piatkowska@urk.edu.pl (E.P.); aneta.kopec@urk.edu.pl (A.K.); teresa.leszczynska@urk.edu.pl (T.L.); 2Department of Plant Biology and Biotechnology, Faculty of Biotechnology and Horticulture, University of Agriculture in Krakow, al. Mickiewicza 21, 31-120 Krakow, Poland; sylwester.smolen@urk.edu.pl

**Keywords:** iodoquinoline, in vivo studies, kale, biofortification, biochemical parameters, iodine deficiency

## Abstract

Background: Iodine is one of the essential trace elements for human life. The main objective of the biofortification of plants with iodine is to obtain food with a higher content of this element compared to conventional food. Biofortification of plants with iodine can increase the intake of this trace element by different populations. In addition, it may reduce the risk of iodine deficiency diseases. Objectives: The aim of the study was to investigate the effect of kale biofortified with 8-hydroxy-7-iodo-5-quinolinesulfonic acid (8-OH-7-I-5QSA) on iodine bioavailability and biochemical effects in Wistar rats. Methods: Kale biofortified with (8-OH-7-I-5QSA) was tested for iodine levels in urine, feces, and selected tissues using the ICP-MS/MS technique. The feeding experiment was designed to investigate potential changes in selected thyroid-regulated biochemical parameters in blood serum of Wistar rats. Results: The dietary intake of Wistar rats fed kale biofortified with (8-OH-7-I-5QSA) from both the “Oldenbor F_1_” and “Redbor F_1_” cultivars for 8 weeks resulted in significantly (*p* ≤ 0.05) higher iodine concentrations in the urine and kidneys of rats, which proves iodine bioavailability. Rats’ diets with “Oldenbor F_1_” and “Redbor F_1_” kale non- and -biofortified with 8-OH-7-I-5QSA had a significantly (*p* ≤ 0.05) lower or a tendency for lower concentration of TSH, triglyceride, total and direct bilirubin, TBARs, uric acid, aspartate aminotransferase (AST) and alanine aminotransferase (ALT) concentrations in serum. Dietary intake of “Oldenbor F_1_” and “Redbor F_1_” kale biofortified with 8-OH-7-I-5QSA significantly (*p* ≤ 0.05) increased the total antioxidant status (TAS). Conclusions: Our study confirms that kale biofortified with iodine in organic form iodoquinoline 8-OH-7-I-5QSA is bioavailable and well absorbed by the Wistar rat and has a positive effect on selected biochemical parameters. The results obtained in this study may be highly predictive for further studies in humans.

## 1. Introduction

Worldwide, around 2,000,000,000 people suffer from micronutrient deficiencies negatively affecting their health. It is estimated that a third of the global population lives in areas where iodine (I) is deficient [1,2,3,4]. Such deficiencies lead to several health problems, including thyroid goiter, reproductive failure, hearing loss, growth disorders, congenital iodine deficiency syndrome and multiple brain injuries. Foetuses are particularly susceptible to iodine deficiency, which can lead to a variety of neurological disorders, ranging from reduced intelligence quotient scores to severe mental retardation (cretinism). Children are susceptible to iodine deficiency at any stage of development [5,6,7].

Iodine is a key trace element required for the production of thyroid gland hormones (3,5,3′-triiodo-L-thyronine, T3, 3,5,3′,5′-tetraiodo-L-thyronine, T4), which are essential for mammalian life [3]. The iodine-replete healthy adult contains about 15–20 mg of iodine, 70–80% of which is contained in the thyroid [8].

Iodine’s bioavailability depends on oral intake, and the recommended daily intake varies according to the age and physiological state of the individual. Iodine can also be absorbed from the air, through the mucous membranes of the respiratory system, and by the skin [2].

The most common method of addressing iodine deficiency is the iodisation of table salt, although critics point out that this does not always cover the full need for the element. Salt iodisation is not sufficient, due to the volatility of iodine and its loss during storage, transport and cooking. In addition, it is not recommended for people with cardiovascular disease, although the impact is still debated [9]. In most iodine-deficient countries, the iodisation of table salt is widely recognised as the most effective way to eliminate iodine-deficiency diseases (IDD). Despite this, one-third of the world’s population is still not protected against iodine deficiency [10].

Mineral deficiencies can be addressed through dietary diversity and supplementation. An alternative and often complementary, method is the of biofortified crops, which can help to address micronutrient deficiencies. Biofortification is the process of increasing the nutritional value of crops through biological techniques such as selective breeding or genetic engineering. The goal of biofortification is to enhance the content of essential vitamins, minerals, and other nutrients in crops to improve the nutritional quality of food, particularly in areas where people may suffer from nutrient deficiencies [11]. Biofortification as a strategy is well documented and economically and environmentally effective in combating mineral malnutrition in various populations [11,12]. Studies on appropriate application methods and dosage of micronutrients in plant crops have shown that this benefits both humans and plants [13,14,15,16,17,18,19,20]. Biofortification aims to enrich plants with nutrients during their growth instead of adding micronutrients after food processing [21]. The most common methods of biofortification include the use of nutrient fertilisers, plant crossing, and genetic manipulation [22,23,24].

Previous research has focused on enriching vegetables with mineral forms of iodine, such as KI and KIO_3_ [20,25,26,27]. Organic iodine compounds such as iodosalicylates and iodobenzoates have been used to enrich lettuce [28,29] and tomatoes [30,31] and produced satisfactory results for iodine accumulation vs. mineral forms. Furthermore, the use of the organic iodine compounds, such as iodosalicylates, fits in with the ecological strategy of improving soil quality and iodine-enriched plants. Moreover, the results of the biofortification of vegetables by iodoquinolines, presented for the first time by our team, showed the effectiveness of these organic forms for the treatment of enriching plants—such as kale, lettuce, and potatoes—with iodine [32,33,34]. Our research team also tested the effects of selected heat treatment on kale biofortified with different forms of iodoquinolines [35,36]. In our study, we used iodoquinolines as an organic form of iodine. This was dictated by the fact that quinoline derivatives have many health-promoting properties, including antioxidant, anti-inflammatory, anticancer, antibacterial, antiviral and antifungal activities [37]. Moreover, we were the first to attempt to biofortify *Brassica oleracea* var. *sabellica* L. with iodoquinolines for several reasons. Kale *Brassica oleracea* var. *sabellica* L. is a rich source of nutrients such as: vitamins (C, A, K), folic acid, minerals (calcium, iron, magnesium) and non-nutrients i.e., polyphenols and glucosinolates [38]. It is indicated that these bioactive substances enhance its anticancer and anti-inflammatory potential [39]. On the other hand, kale as a cruciferous vegetable, contains substances that impede iodine absorption, therefore we assumed that iodoquinolines combined with kale—a vegetable with proven health-promoting properties—would bring tangible health benefits: (1) additional iodine in the diet and (2) contribute to increased consumption of kale, which will translate into health benefits for consumers.

Wistar rats are commonly used for nutritional experiments due to several reasons. Firstly, rodents are easily available and have been historically used in laboratories due to their prolificacy, robustness, and tameness. Secondly, the small intestine of rats has a similar histological structure to that of humans, making them suitable models for various experimental studies. Overall, using rats in experiments allows researchers to study multiple aspects of physiology, toxicology, and disease mechanisms. Therefore, our study used rats to evaluate the bioavailability of iodine from iodoquinolines derived from kale.

In our study, to our knowledge, we were the first to show that that kale *Brassica oleracea* var. *sabellica* L. biofortified with iodine in organic form 8-OH-7-I-5QSA is bioavailable and well absorbed by the Wistar rat and has a positive effect on selected thyroid-regulated biochemical parameters.

## 2. Materials and Methods

### 2.1. Analysis of Plant Material

Kale “Oldenbor F_1_” and “Redbor F_1_” cultivation and biofortification with 8-hydroxy-7-iodo-5-quinolinesulfonic acid has been previously described [36]. The process of preparation of the initial test material, i.e., kale *Brassica oleracea* var. *sabellica* L., basic chemical composition and iodine analysis, was described in an earlier publication by Krzemińska et al., 2024 [36] using AOAC standards [40,41,42]. The samples of kale were analysed to determine the concentrations of iodine, using the ICP-MS/MS technique. The analysis was based on research published by Smoleń et al. (PN-EN, 15111:2008) [43,44].

### 2.2. Animal Study

Five-week-old—albino Wistar wandering rats (n = 40, males) were obtained from the Animal Husbandry at the Animal Jagiellonian University Medical College, Krakow, Poland. The first Local Ethical Committee approved the experimental procedures for the Animal Experiments in Krakow (Poland, res. no. 568/2021, date 28 October 2021, Jagiellonian University—Faculty of Pharmacy ul. Medyczna 9, 30-688 Krakow). Before the experiment, the animals were acclimatised for 7 days on standard laboratory chow. At the end of the acclimatisation period, the animals were randomly divided into five experimental groups (n = 8). The mean body weight of the rats at the beginning of the experiment was 132 ± 10 g. The experimental diets were prepared based on AIN-93G [45]. The detailed composition of the diets can be found in Table 1, below. The research on iodine bioavailability was carried out only on male rats due to the limited number of animals in the study (according to the 3R rule). Randomization was performed by randomly assigning experimental rats to the experimental group and the control group.

Group 1 was fed an AIN-93G (C) diet, consisting of a mineral mixture containing iodine as recommended by [45]. For Group 2 (CO) with control “‘Oldenbor F_1_” kale and for Group 4 (CR) with control “Redbor F_1_” kale, both diets were prepared with a mineral mixture with iodine. In the diets containing biofortified kale (Group 3, BO diet with biofortified 8-OH-7-I-5QSA “Oldenbor F_1_” kale and Group 5, BR diet with biofortified 8-OH-7-I-5QSA “Redbor F_1_” kale), the only source of iodine was kale (the mineral mixture was devoid of iodine—Table 1). Rodents were individually housed in steel metabolic cages during the first and eighth weeks of the experiment. The temperature was maintained at 21 °C and the light/dark cycle was 12/12 h. The animals had unlimited access to deionised distilled water throughout the experiment. Use of distilled rather than tap water to limit rats’ intake of additional iodine. The intake of the experimental diets was recorded daily (8:00–9:00 in the morning). During the remaining weeks, rats were housed in conventional open-air cages with two or three individuals per cage. The cages were equipped with elements that enrich the environment, such as wooden blocks/nest material, as well as tunnels that serve to activate the animals. The cages were equipped with houses/platforms that provide shelter, which in turn reduces anxiety and possibly aggressive behaviour of rats. Moreover, marking involving the microchipping, ear-cutting, or ear-tagging of rodents causes additional stress and pain to the animals, which in this case are not necessary. Body weight gain was measured weekly for the entire duration of the experiment, which lasted 8 weeks. Data on the amount of feed consumed and weight gain of the rats were used to calculate the FER—feed efficiency ratio—(body weight gain (g)/diet consumed (g)). Results were expressed as a fresh weight (µg·L^−1^) in urine and a dry weight (g·kg^−1^) in faeces.

Urine and faeces were collected from Day 1 to Day 5 and from Day 49 to 54 of the experiment (Week 1 and Week 8, respectively) to assess iodine excretion. The collected samples were stored at −20 °C until analysis. At the end of the 8-week experimental period, fasted rats (12 h) were anaesthetised (isoflurane 4%, inhaled, AErrane Isoflurane USP, Baxter Corporation Mississauga, ON, Canada). Humane endpoints used in the study were administration of an analgesic (Morphasol, LIVISTO St. Chwaszczyńska 198a. 81-571 Gdynia, Poland) and euthanasia by anaesthetic overdose (isoflurane). Monitored symptoms: euthanasia of rats. After euthanasia, the animals will be transferred to the Sanitary Plants in Krakow, LLC, ul. Dymarek 7, 31-983 Krakow. Blood was collected by cardiac puncture into standard tubes. Blood samples were centrifuged (1500× *g*, 15 min) to obtain serum. The liver, kidney, thyroid gland, and heart were excised, washed with 0.9% sodium chloride solution, dried with a paper towel, and weighed. Serum and tissue samples were frozen at −80 °C for analysis.

### 2.3. Iodine Content in Urine, Faeces, and Selected Tissues

Collected samples of urine were adjusted to the same volume before analysis. The faeces, kidneys, livers were freeze-dried. After freeze-drying, the organs were weighed and ground in a laboratory hurricane mill (WZ-1. Sadkiewicz Instruments. ZDPP. Bydgoszcz, Poland). Then, the prepared samples (particle size about 1 mm) were used for measurements of iodine content. The content of iodine in these samples was analysed using the ICP-MS/MS technique (Inductively coupled plasma triple quadrupole mass spectrometry; iCAP TQ ICP-MS; ThermoFisher Scientific, Bremen, Germany), (PN-EN. 15111:2008) [44]. The procedure for the analysis of iodine in faeces, kidneys, and livers was the same as for plant samples. Urinary iodine content was analysed according to the following procedure:

Urine samples were thawed and mixed. Then 4.8 mL of urine was collected into a PP tube, 0.2 mL of 25% TMAHu was added and incubated with this reagent. Subsequently, the samples were mixed and diluted 200× in water redistilled three times. Then, the analysis was performed using the ICP-MS/MS technique.

### 2.4. Analysis of Serum

Serum levels of total cholesterol—TC; (cat. no. Liquick Cor-CHOL 60, 2-204, PZ Cormay S.A., Lublin, Poland), HDL cholesterol (cat. no. CORMAY HDL, 2-053, PZ Cormay S.A.) and triglycerides-TG (cat. no. Liquick Cor-TG 60, 2-253, PZ Cormay S.A.) were measured. Differences between TC and HDL were used to calculate LDL + VLDL levels [46]. The liver panel was examined including ALAT alanine aminotransferase activity; (cat. no. Liquick Cor-ALAT 60, 1-217, PZ Cormay S.A.), aspartate aminotransferase ASAT (cat. no. Liquick Cor-ASAT 60, 1-214, PZ Cormay S.A.). Total bilirubin—TB (cat. no. Liquick Cor-BIL TOTAL 60, 2-245, PZ Cormay S.A.), and direct bilirubin—DB (on cat. Liquick Cor-BIL DIRECT MALLOY-EVELYN 60, 2-348, PZ Cormay S.A.) were determined. Serum uric acid levels were also tested (cat. no. Liquick Cor-UA 60, 2-208, PZ Cormay S.A.).

Triiodothyronine (T3) levels were measured using ELISA Tests Enzyme-linked Immunosorbent Assay Kit for Triiodothyronine (cat. no. CEA453Ge, Cloud-Clone Corp, Katy, TX, USA), and thyroxine (T4) levels were measured using ELISA Tests Enzyme-linked Immunosorbent Assay Kit for Thyroxine (cat. no. CEA452Ge, Cloud-Clone Corp, Katy, TX, USA). Thyrotropic hormone (TSH) concentrations were determined with the Rat Thyroid Stimulating Hormone ELISA kit (cat. no. RTC007R, BioVendor R&D, Karásek, Brno, Czech Republic).

Glutathione reductase levels were determined using the Glutathione Reductase (GLUT RED) kit (cat. no. GR 2368, RANDOX, Crumlin, UK) and total oxidant status using the Total Antioxidant Status (TAS) kit (cat. no. NX 2332, RANDOX, Crumlin, UK). Determination of serum lipid peroxidation products was performed using the thiobarbituric acid—TBARs interaction according to Ohkawa, Ohishi, and Yagi [47]. Results were expressed in terms of free malondialdehyde (MDA) a marker of lipid peroxidation, which is a marker of oxidative stress.

### 2.5. Statistical Analysis

The data were presented as mean ± SD (n = 8). The experimental data were analysed using a one-factor analysis of variance (ANOVA), except for the content of iodine in urine and faeces (two-factor). Statistics were performed using the Statistica software v. 13.1 PL (Dell Inc., Tulsa, OK, USA). The level of significance considered was *p* ≤ 0.05. In the case of significant effects, homogeneous groups were distinguished by Duncan’s post hoc test. In the two-factor experiment, the variables were: Factor No. 1 type of diet: C—control diet (AIN-93G), CO—diet containing control curly kale *Brassica oleracea* var. *sabellica* L. “Oldenbor F_1_”, BO—diet containing biofortified curly kale *Brassica oleracea* var. *sabellica* L. “Oldenbor F_1_”, CR—diet containing control curly kale *Brassica oleracea* var. *sabellica* L. “Redbor F_1_”, BR—diet containing biofortified curly kale “Redbor F_1_”, x Factor No. 2 week: I, VIII.

## 3. Results

### 3.1. Iodine Content and Basic Chemical Composition in Plant Material

Research into the basic chemical composition of kale and a description of the results were necessary to compose an appropriately balanced diet for the presented in vivo experiment.

The control (non-biofortified) kale *Brassica oleracea* var. *sabellica* L. of both cultivars “Oldenbor F_1_” and “Redbor F_1_” was not significantly different in terms of iodine content (Table 2). Application of the biofortification process with 8-OH-7-I-5QSA significantly increased the iodine content of the plants of both cultivars, from 0.18 to 2.10 of “Oldenbor F_1_” and from 0.20 to 2.43 of “Redbor F_1_” mg·kg d.m.^−1^, respectively. Furthermore, the red cultivar “Redbor F_1_” accumulated a significantly higher iodine content (*p* ≤ 0.05).

Analysing the results in Table 2 showed that green kale “Oldenbor F_1_” (control and biofortified with 8-OH-7-I-5QSA) had more protein (*p* ≤ 0.05) than kale of the red cultivar “Redbor F_1_” (control and biofortified with 8-OH-7-I-5QSA).

The total fat content of green kale of the “Oldenbor F_1_” cultivar in the control as well as the biofortified version was also higher (*p* ≤ 0.05) than that of red kale of the “Redbor F_1_” cultivar in the control and biofortified version.

For digestible carbohydrates, we notice significant changes (*p* ≤ 0.05) after biofortification with 8-OH-7-I-5QSA. For both cultivars of kale “Oldenbor F_1_” and “Redbor F_1_”, the biofortification process with 8-OH-7-I-5QSA increased the digestible carbohydrate content compared to the control. Significant differences in digestible carbohydrates were observed between the two cultivars. The green cultivar “Oldenbor F_1_” had more digestible carbohydrates than the red cultivar “Redbor F_1_”.

The ash content of “Redbor F_1_” kale was higher (*p* ≤ 0.05) compared to “Oldenbor F_1_”, for both control and biofortified with 8-OH-7-I-5QSA kale. A decreasing effect of the 8-OH-7-I-5QSA biofortification process, for both cultivars of kale ‘Oldenbor F_1_’ and “Redbor F_1_”, on the ash content was observed (*p* ≤ 0.05). The differences in ash content between kale cultivars such as “Redbor F_1_” and “Oldenbor F_1_” are due to several factors, mainly related to their genetics, plant structure, and growing conditions. More information is included in the discussion.

The kale of the red cultivar “Redbor F_1_” had a higher dietary fibre content compared to the green cultivar “Oldenbor F_1_” (*p* ≤ 0.05). “Redbor F_1_” kale cultivar has more fibre than “Oldenbor F_1_” probably due to its thicker, crimped leaves, stress resistance, and genetic differences in the cell wall structure. Breeding selection for leaf thickness and texture in “Redbor F_1_” leads to a higher fibre content, which distinguishes it from the more tender “Oldenbor F_1_” cultivar. Differences in fibre content are described in the discussion.

### 3.2. Body Weight Gain, Weight of Selected Organs

Table 3 shows the set of results on body gain, feed efficiency ratio (FER) and weight of selected organs. Dietary intake of Wistar rats of non-biofortified kale *Brassica oleracea* var. *sabellica* L. (CR) and kale biofortified with 8-OH-7-I-5QSA of both cultivars, “Oldenbor F_1_” (BO) for 8 weeks resulted in significantly lower body weight gain compared to the AIN-93G (C) control. A trend towards lower body weight gain was shown by the CO, BR groups.

The feed efficiency ratio (FER) of rats fed diets with non-biofortified kale (CO, CR) and kale biofortified with 8-OH-7-I-5QSA (BO, BR) was lower compared to the fed control diet AIN-93G (C) without kale, but only significantly so (*p* ≤ 0.05) for the BO and CR groups.

The liver weights of rats fed non-biofortified kale (CO, CR) and biofortified with 8-OH-7-I-5QSA kale (BO, BR) diet were the lowest (*p* ≤ 0.05) compared to the AIN-93G (C).

The sum of the weights of both kidneys in the group of rats fed the control diet AIN-93G (C) was similar (*p* > 0.05) compared to the other experimental groups (CO, BO, BR). Kidneys weight was significantly lowest in the group of rats that fed non-biofortified “Redbor F_1_” kale (CR). Lower kidney and liver weights in Wistar rats fed non-biofortified kale (without iodoquinoline) may be due to several physiological and biochemical factors. This is described in more detail in Section 4.

Heart weight in the group of rats fed the AIN-93G (C) control diet was similar too (*p* > 0.05) compared to the other experimental groups fed non-biofortified and biofortified with 8-OH-7-I-5QSA kale (CO, BO, BR). Significantly decrease of heart weight was only for CR group.

The thyroid gland weight of rats fed a diet enriched with non-biofortified kale (CO, CR) and kale biofortified with 8-OH-7-I-5QSA (BO) was similar (*p* > 0.05) compared to control group (C), except for the BR group, where the weight of this organ increased significantly.

Visceral fat content was not significantly different in any group of rats fed non-biofortified kale (CO, CR) and kale biofortified with 8-OH-7-I-5QSA of both cultivars (BO, BR). In our study, visceral fat content did not differ significantly in any rat study group (*p* > 0.05).

### 3.3. Iodine Excretion in Urine, Faeces, and Selected Organs

The bioavailability of iodine from the gastrointestinal tract of Wistar rats in the presented experiment was controlled by iodine excretion with urine and faeces. After 8 weeks on the diet, higher urinary iodine concentrations were determined in the BO study group. In addition, a trend towards higher urinary iodine concentrations after 8 weeks was also shown in the BR group. Thus, feeding diets with kale *Brassica oleracea* var. *sabellica* L. of both cultivars (“Oldenbor F_1_” and “Redbor F_1_”) biofortified with 8-OH-7-I-5QSA generally increased urinary iodine concentrations in time (Figure 1A).

A different situation was observed in the faecal iodine content of the Wistar rats tested. In general, no significant changes were shown in faecal iodine content at the beginning and end of the experiment (Figure 1B).

The highest kidney iodine concentrations were determined in rodents fed diets of biofortified kale *Brassica oleracea* var. *sabellica* L. cultivars “Oldenbor F_1_” (BO) and “Redbor F_1_” (BR). Biofortified “Oldenbor F_1_” kale (BO) had a higher iodine content in kidneys, than non-biofortified kale (CO) Figure 2A.

The same situation was for biofortified “Redbor F_1_” kale (BR), where kidney iodine content was higher than non-biofortified control kale (CR). No similar results were found in the livers of the experimental rats (Figure 2B).

### 3.4. Selected Biochemical Parameters

The effect of using iodoquinoline (an organic form of iodine) in the diet of Wistar rats had to be controlled by analysing selected thyroid-regulated biochemical parameters. The feeding of different types of diets by Wistar rats did affect the change (*p* > 0.05) in the activity of the liver enzyme aspartate aminotransferase (AST) after 8 weeks of experimentation (Table 4). After feed of BR kale, a decrease (*p* > 0.05) in AST enzyme activity or a tendency for BO to decrease (*p* > 0.05) was observed. Feeding of the kale *Brassica oleracea* var. *sabellica* L. cultivar “Redbor F_1_” reduces AST enzyme concentrations the most. In the liver enzyme alanine aminotransferase (ALT), where dietary use of “Redbor F_1_” kale non-biofortified (CR) and “Redbor F_1_” biofortified with 8-OH-7-I-5QSA (BR), resulted in a significant reduction in enzyme activity or only a tendency for it (*p* > 0.05), in the case of “Oldenbor F_1_” kale enriched in 8-OH-7-I-5QSA (BO). The addition of “Redbor F_1_” non-biofortified and biofortified red kale to the diets significantly affected lower enzyme activity values, compared to the AIN-93G (C) diet and “Oldenbor F_1_” control green kale (CO).

Total bilirubin values were the highest (*p* ≤ 0.05) in the group of rats fed the AIN-93G (C) control diet. The other experimental groups that were fed non-biofortified and biofortified kale had significantly lower total bilirubin values (Table 4). A similar situation was observed for direct bilirubin, where the experimental groups (CO, BO, BR), compared to the AIN-93G diet (C), showed lower concentrations (*p* ≤ 0.05) or a tendency (*p* > 0.05) for lower concentration values (CR).

The serum uric acid values of Wistar rats in our experiment were not affected by the diets (*p* > 0.05). However, there was a trend towards lower uric acid concentrations in Wistar rats after feeding non- and biofortified with 8-OH-7-I-5QSA kale in both cultivars. Moreover, the lowest uric acid concentrations were in the groups where iodoquinoline was added. Iodoquinol, which is a component of the tested compound 8-OH-7-I-5QSA, may contribute to the reduction of serum uric acid concentration in Wistar rats through various mechanisms. The mechanism of action of iodoquinol on uric acid concentrations is described in more detail in the Section 4.

Total cholesterol (TC) values in rats fed different diets did not differ significantly between groups (Table 4). HDL fraction cholesterol values were similar in all groups. Only a tendency for the highest HDL value was recorded in the AIN-93G (C) fed rats compared to the CO, BO and BR groups (*p* > 0.05). However, a significantly lower HDL fraction cholesterol value was recorded in the group of rats fed “Redbor F_1_” control kale (CR) compared to the C group. LDL + VLDL cholesterol concentrations were not significantly different between all experimental groups. Triglyceride concentrations had only a tendency for the highest concentration in the group of rats fed AIN-93G (C), compared to the other experimental groups (CO, CR, BR), or were significantly higher compared to the BO group.

Antioxidant activity was determined in the blood serum of the test rats by glutathione reductase (RG), total antioxidant status (TAS) and products of lipid peroxidation by reaction with thiobarbituric acid (TBARs)—Table 4. After 8 weeks of the experiment, glutathione reductase levels did not differ significantly between the experimental groups (*p* > 0.05).

Serum TAS determination in rats showed that dietary intake of “Oldenbor F_1_” (BO) and “Redbor F_1_” (BR) kale biofortified with 8-OH-7-I-5QSA significantly increased (or in one case resulted in a tendency) total antioxidant status relative to dietary intake of AIN-93G (C) and “Oldenbor F_1_” green kale (CO).

The concentration of thiobarbituric acid reactive substances (TBARs) expressed in nmol malondialdehyde (MDA)·mL^−1^ was significantly highest in the AIN-93G control group (C) in comparison to the group fed “Redbor F_1_” (CR, BR), and a tendency for the highest concentration was found in comparison with the CO and BO groups (*p* > 0.05). MDA concentrations were dependent on the use of non-biofortified and biofortified “Redbor F_1_” (CR, BR). It was observed that the addition to the diet of these two kinds of kale significantly reduced MDA concentrations in rats compared to rats fed with the control (C) diet (Table 4).

The highest thyrotropic hormone (TSH) concentrations were observed in the experimental group feeding the AIN-93G (C) diet compared to the groups fed the CO and BO diet (*p* ≤ 0.05) or a tendency for the highest concentration was found compared to the groups fed the CR and BR diet (*p* > 0.05). Triiodothyronine (T3) and thyroxine (T4) hormone concentrations were not significantly different between all experimental groups of rats tested (*p* > 0.05).

## 4. Discussion

Iodine is one of the trace elements essential for human life. The main objective of the biofortification of plants with iodine is to obtain foods with a higher content of this element compared to conventional foods. Biofortification of plants with iodine can increase the intake of this trace element by different populations. In addition, it may reduce the risk of iodine deficiency diseases [48].

In this study, the effects of kale (*Brassica oleracea* var. *sabellica* L.) of two cultivars, “Oldenbor F_1_” and “Redbor F_1_” biofortified with iodine in the form of 8-OH-7-I-5QSA, on iodine content in urine, faeces, selected tissues, and various biochemical parameters of Wistar rats were evaluated for the first time.

Before moving on to the analysis of the in vivo experiment, we will first focus on the evaluation of iodine content and the basic chemical composition of the test kale *Brassica oleracea* var. *sabellica* L. of two cultivars, “Oldenbor F_1_” and “Redbor F_1_”, which was added to the diet of Wistar rats. Kale is a long-known vegetable with high nutritional value due to its high content of bioactive compounds and macro- and micronutrients [49]. Improving the nutritional value of this vegetable by increasing the overall iodine content, through the application of a biofortification process, further enhances this crop. Application of iodine in the form of 8-OH-7-I-5QSA by nutrient solutions in hydroponic cultivation significantly increased the iodine content and did not affect the basic chemical composition of kale compared to control plants. However, differences in basic chemical composition were observed depending on the cultivar tested.

Application of the biofortification process with 8-OH-7-I-5QSA significantly increased the iodine content of the plants from 0.18 to 2.10 mg·kg s.m.^−1^ in the “Oldenbor F_1_” cultivar and from 0.20 to 2.43 mg·kg s.m.^−1^ in the “Redbor F_1_” cultivar, respectively. Furthermore, the red cultivar “Redbor F_1_” accumulated a significantly higher iodine content (2.43 mg·kg d.m.^−1^) after biofortification compared to the green cultivar “Oldenbor F_1_” (2.10 mg·kg d.m.^−1^). The kale “Redbor F_1_” of cultivar can accumulate more iodine than other cultivars, such as “Oldenbor F_1_”, due to several key factors related to genetics, leaf structure and the plant’s ability to absorb minerals from the soil. Kale cultivars can vary in the structure and efficiency of their root system, which affects their ability to absorb minerals, including iodine, from the soil. The “Redbor F_1_” cultivar may have a more developed root system better adapted to efficiently take up iodine from the soil. The “Redbor F_1_” genotype may influence the higher activity of enzymes and proteins responsible for the transport and storage of iodine in plant cells. Each kale cultivar is genetically different and “Redbor F_1_” may be more predisposed to accumulate higher amounts of iodine. This requires further research. The “Redbor F_1_” cultivar is rich in anthocyanins (phenolic compounds) [50], which may affect the absorption and transport of minerals in the plant. These compounds can promote the storage of trace elements, including iodine, leading to greater iodine accumulation compared to varieties with lower anthocyanin content, such as “Oldenbor F_1_”. Anthocyanins can protect the plant against oxidative stress and UV radiation [50], which can improve the efficiency of mineral uptake, including iodine, from the soil. The “Redbor F_1_” with its more developed, thicker leaves, can better manage transpiration, which promotes more efficient transport of iodine to the leaves. Previous studies by many authors confirm the effectiveness of the iodine biofortification process in increasing the content of this element in vegetables. These studies have mainly focused on enriching of vegetables with mineral forms of iodine (KI and KIO_3_) [20,25,26,27]. Recently, our team has begun research on enriching plants, including vegetables, with organic forms of iodine. Organic forms of iodine, i.e., iodosalicylates and iodobenzoates, have been used to enrich—for example—lettuce [28,29] and tomatoes [30,31]. Moreover, the results of biofortification of vegetables by iodoquinolines, presented for the first time by our team, showed the effectiveness of these organic forms for the treatment of enriching plants—such as kale, lettuce, and potatoes—with iodine [32,33,34].

Analysing the results of our study, it was observed that green kale “Oldenbor F_1_” (control and biofortified with 8-OH-7-I-5QSA) contained more protein than kale (control and biofortified with 8-OH-7-I-5QSA) of the red cultivar “Redbor F_1_”. Green “Oldenbor F_1_” control kale had 32.98 g·100 g d.m.^−1^ of protein and biofortified 8-OH-7-I-5QSA had 30.74 g·100 g d.m.^−1^, respectively. In contrast, red “Redbor F1” control kale had less protein at 26.03 g·100 g d.m.^−1^ and biofortified 8-OH-7-I-5QSA at 25.71 g·100 g d.m.^−1^, respectively. Slightly lower results were obtained in the work of Prade et al. [51], where the protein content of kale leaves of the cultivar *Brassica oleracea* var. *sabellica* L. was 150 g·kg d.m.^−1^. In the work, Pitura and Jarosz [52] showed that the total protein content in the dry matter of kale of another medium-high cultivar (*Brassica oleracea* var. *acephala* L.) ranged from 243.7 to 350.6 g·kg d.m.^−1^. In all cases, the values did not differ significantly, but in general, the protein content of the plant is dependent on the crop year, harvest date and cultivar [53].

The total fat content of green kale of the control and biofortified version of the “Oldenbor F_1_” cultivar was higher than that of red kale of the control and biofortified version of the “Redbor F_1_” cultivar. Kale, like most vegetables, has a low total fat content. In our study, the fat content was in the range of 3.16–5.46 g·100 g d.m.^−1^. According to Satheesh and Fanta [38], the total fat content of kale *(Brassica oleracea* var. *acephala* L.) was higher at 11.8 g·100 g d.m.^−1^. The differences in fat content, as with total protein content, may be a result of crop year, harvest date, and cultivar [53].

Significant differences in digestible carbohydrates were observed between the two cultivars. The green cultivar “Oldenbor F_1_” had more digestible carbohydrates than the red cultivar “Redbor F_1_”. For digestible carbohydrates, we noticed significant changes after biofortification with 8-OH-7-I-5QSA. For both cultivars of kale, “Oldenbor F_1_” and “Redbor F_1_”, the biofortification process with 8-OH-7-I-5QSA increased the digestible carbohydrate content compared to the control while increasing the nutritional value. Overall, carbohydrate content ranged from 15.53-20.39 g·100 g d.m.^−1^.

Our study showed that kale of the red cultivar “Redbor F_1_” had a higher dietary fibre content compared to the green cultivar “Oldenbor F_1_”. Dietary fibre is important in the prevention of many diseases. Higher intake has a positive effect on improving insulin sensitivity, modulating the secretion of certain gut hormones and influencing various metabolic and inflammatory markers associated with the metabolic syndrome [54]. This makes the red cultivar “Redbor F_1_” more attractive to consumers in this respect. The kale cultivar “Redbor F_1_” is characterised by more robust, often wavy and thicker leaves. This leaf structure may be due to the higher content of cellulose and hemicellulose, which are the main fibre components. The genes responsible for cell wall thickness in “Redbor F_1_” may be more active than in “Oldenbor F_1_”, leading to higher fibre production. “Redbor F_1_” kale has a dark purple colour, indicating the presence of anthocyanins [50], which can affect the structure and hardness of the leaves. An increase in lignin, which is also part of the fibre, may occur to strengthen the cell walls and protect the plant, increasing the total amount of fibre. “Redbor F_1_” kale often has crimped leaves, which may contain more structural tissue. This crimped texture suggests more structural fibre compared to the “Oldenbor F_1_” cultivar, which has a smoother texture. The “Redbor F_1_” cultivar is often more tolerant of harsh environmental conditions such as cold or drought, which may lead to increased production of lignin (a fibre component) in response to these stresses.

The ash content of “Redbor F_1_” kale was higher than “Oldenbor F_1_” kale. It was observed that the biofortification process with 8-OH-7-I-5QSA reduced the ash content of both cultivars of kale “Oldenbor F_1_” and “Redbor F_1_”. Ash content is the amount of minerals remaining after the organic matter has been burnt. The “Redbor F_1_” cultivar may have more efficient mechanisms for the uptake of nutrients such as calcium, potassium, magnesium or iron, leading to a higher ash content after burning. The dark purple color of “Redbor F_1_” is due to its higher content of anthocyanins [50], which are mineral-rich compounds. Anthocyanins can promote a higher accumulation of certain minerals, which ultimately increases the ash content. “Redbor F_1_”, compared to “Oldenbor F_1_”, tends to have thicker and fleshy leaves, which may contain more nutrients and minerals, increasing the overall ash content. “Redbor F_1_” may be more efficient at taking up minerals from the soil compared to “Oldenbor F_1_”, which may be due to differences in genes, root system structure or soil preference. “Redbor F_1_” cultivar may have a thicker waxy coating on the leaves, which may contain minerals and contribute to higher ash content after burning. This coating has a protective function and, at the same time, may accumulate more minerals.

Feeding by Wistar rats of diets supplemented with non-biofortified (CO, CR) and biofortified 8-OH-7-I-5QSA (BO, BR) kale *Brassica oleracea* var. *sabellica* L. of both “Oldenbor F_1_” and “Redbor F_1_” cultivars for over 8 weeks resulted in significantly lower animal weight gain values compared to the control diet AIN-93G (C)—without added kale. Similarly, the feed efficiency ratio (FER) of rats fed diets with non-biofortified kale (CO, CR) and kale biofortified with 8-OH-7-I-5QSA (BO, BR) was lower compared to the AIN-93G (C) control diet without kale, but significantly (*p* ≤ 0.05) only for the BO and CR groups. Adding freeze-dried kale to the diets reduced the excessive weight gain of the test individuals. This may be due to the presence of high amounts of dietary fibre and many other bioactive compounds (polyphenols, glucosinolates) found in kale. Although we know that in our study the visceral fat content did not differ significantly in any rat study group, the highest value was observed in the AIN-93G (C) group at 4.39 g. Bioactive compounds reduce the absorption of lipids from the gastrointestinal tract and thus affect lower body weight gain in Wistar rats. In a study by Kopeć et al. [48], a diet with biofortified lettuce in the KI did not affect the weight gain values of individuals and the feed efficiency ratio FER. However, a study by Piątkowska et al. [55] showed that of KI-enriched cooked carrots by Wistar rats resulted in higher weight gain of individuals, compared to the AIN-93G diet. In a study by Rakoczy et al. [56], fed of lettuce biofortified with KI did not affect body weight gain or feed conversion ratio FER.

Diets with added kale *Brassica oleracea* var. *sabellica* L. non- and -biofortified with 8-OH-7-I-5QSA of both cultivars, “Oldenbor F_1_” and “Redbor F_1_”, had a significant effect on lower liver weight. Significant reductions in kidney and heart weight were observed in a diet supplemented with non-biofortified “Redbor F_1_” kale. The use of iodoquinolines influenced lower values of total body weight in the rats and thus the weight of individual organs. It can be suggested that the presence of various bioactive compounds in kale, especially polyphenolic compounds and dietary fibre, may result in lower absorption of crude fat from the gastrointestinal tract, which may have influenced the lower weight of the above organs. In a study by Kopeć et al. [48], heart and kidney weights in rats after feeding lettuce biofortified with KI did not change. However, liver weight was significantly higher (*p* ≤ 0.05) in rats fed the AIN-93G control diet without added lettuce, compared to the liver of rats fed the diet with KI—biofortified lettuce. In a study by Rakoczy et al. [56], the addition of (non-) biofortified lettuce (KI) to the diet did not affect the kidney, liver, heart, or femoral muscle weights of rats. In a study by Piątkowska et al. [55], the feeding of KI-biofortified raw and cooked carrots influenced higher liver weight. In the same study, kidney weight was not affected by the different dietary treatments and the highest heart weight was found in groups of rats fed the AIN-93G (C) diet compared to the other experimental groups. It can be postulated that depending on the experiment, the used dietary treatments affected the weight of individual organs differently. This is influenced by several factors. e.g., the type of vegetable used in the feed, the content of bioactive substances in the feed, the level and the form of iodine or the growing conditions. Non-biofortified kale (Group C) may contain fewer essential minerals and vitamins (such as iron, zinc, and iodine), which are important for normal kidney function and growth. Biofortification usually increases the nutritional value of plants, including minerals as confirmed by the Krawczyk et al. (2024) kale study [34]. The higher mineral content of biofortified kale may influence the greater growth of this organ. Non-biofortified kale (Group C) may contain fewer antioxidants or other bioactive compounds that help alleviate oxidative stress. Without these compounds, the kidneys of rats consuming non-biofortified kale (group C) may be more susceptible to damage and cellular stress, leading to less organ weight. Kale that is non-biofortified may have a reduced ability to detoxify harmful substances. If a diet with non-biofortified kale (Group C) has a lower content of detoxifying compounds, this may increase the metabolic load on the kidneys, causing renal stress and potential atrophy, which contributes to reduced kidney mass.

The thyroid gland weight of rats fed diets with added kale *Brassica oleracea* var. *sabellica* L. (CO, BO, CR) was similar compared to the AIN-93G control diet—without added kale (C). Only the thyroid gland weight of rats with added kale BR was significantly higher compared to the AIN-93G control diet and other groups. The increase or decrease in organ weight depending on the provision of adequate iodine with the diet depends on several factors. For example, in a study by Sherrer et al. [57] rats were given 0, 1, 3, 10, and 100 mg·L^−1^ of iodine or iodide (in the form of Nal) in their drinking water for 100 days. Thyroid gland weight in male rats increased significantly with increasing iodide concentration in the water. In contrast, thyroid gland weight in female rats decreased after the highest dose of iodide was administered. The results of this study indicate that iodine and iodide affect thyroid gland weight in fundamentally different ways in individuals of different sexes. In our study, the iodine level was the same in each experimental group. Thus, it was not the iodine dose that influenced the increase in organ weight. Perhaps, the differences were due to the origin of the iodine. The AIN-93G diet used iodine from a mineral mixture, the CO, CR diets used iodine found in green and red kale of natural origin and iodine of organic origin (BO, BR) after hydroponic application of 8-OH-7-I-5QSA. In addition, the increase in organ size may have been influenced by components found in kale e.g., goitrogens (goitrogenic substances). These are anti-nutritional compounds naturally occurring in foods (cabbage, broccoli, kale, etc.) that reduce the bioavailability of iodine from food or interfere with the absorption of iodine by the thyroid gland and consequently the production of its hormones. The pituitary gland, in response to a decrease in the concentration of hormones produced by the thyroid gland, releases thyroid-stimulating hormone (TSH), which causes excessive proliferation of thyroid gland tissue, ultimately leading to thyroid goitre [58].

One of the diagnostic methods for iodine deficiency (ID) used in this study was the determination of urinary iodine concentration. The assessment of human iodine nutrition is based on urinary iodine excretion [59] whereas faecal iodine concentration is not relevant for the assessment of iodine nutrition. However, iodine nutrition in rats can be assessed based on iodine content in urine, faeces and organs. Kirchgessner, He, and Windisch [60] observed that iodine concentrations in urine, faeces and organs of rats increased with increasing iodine content in the diet.

The highest urinary iodine excretion was measured at week eight in groups fed diets supplemented with 8-OH-7-I-5QSA biofortified kale “Oldenbor F_1_” (BO) and “Redbor F_1_” (BR) compared to the other experimental groups (C, CO, CR). A different situation was observed in the faecal iodine content of the Wistar rats tested. In general, no significant changes were shown in faecal iodine content at the beginning and the end of the experiment. However, it is noteworthy that there was a trend towards higher faecal iodine concentrations after 8 weeks in the diets with kale biofortified with 8-OH-7-I-5QSA (BO, BR). It is known that iodine is absorbed in the stomach and duodenum and removed by the kidneys and thyroid gland. In our experiment, significantly higher iodine concentrations after 8 weeks of the experiment were found in the kidneys of rats fed a diet containing biofortified 8-OH-7-I-5QSA kale of the “Oldenbor F_1_” as well as the “Redbor F_1_” cultivars compared to the AIN-93G (C) and CO and CR diets. Only for the liver of the BO group was there a tendency for an increase in iodine concentration—(*p* > 0.05). Increased iodine accumulation in the kidney and liver following diets with KI-biofortified vegetable additives is confirmed by studies by Kopeć et al. [48], Piatkowska et al. [55], and Rakoczy et al. [56]. When iodine in the diet is present as I-, it is rapidly and efficiently absorbed in the gastrointestinal tract [61]. Our study showed that iodine contained in biofortified kale *Brassica oleracea* var. *sabellica* L. was more bioavailable. Rats fed diets supplemented with biofortified 8-OH-7-I-5QSA kale (BO, BR) accumulated more iodine in selected organs e.g., kidneys. In our study, iodine doses were equal in each experimental group, confirming that the high bioavailability of the element in tissues is a result of the origin of the iodine, i.e., the organic form, and not the excessive amount of iodine supplied with the diet. Despite the increased excretion of iodine with urine after 8 weeks of the experiment, further high tissue saturation with iodine was achieved. It can therefore be suggested that the accumulation of iodine in the organs is probably to protect the animals from the potential occurrence of ID in food. Furthermore, in the groups with low dietary iodine bioavailability C, CO, and CR urinary and faecal iodine excretion were lower after 8 weeks to protect the organism from iodine deficiency.

The diagnosis of ID is based on the analysis of urinary iodine concentration, TSH concentration and the development of thyroid gland goitre [59]. Two methods of ID diagnosis were used in this study. The first was the analysis of urinary iodine levels in Wistar rats, which has been discussed above. The second method was the examination of thyrotropic hormone, which is a biochemical marker of thyroid gland function and its hormones. TSH increases iodine uptake by the thyroid gland and stimulates the secretion of the hormones thyroxine (T4) and triiodothyronine (T3) into the blood [62]. Elevated TSH levels may be indicative of impaired T4 and T3 synthesis and insufficient dietary iodine supply. The highest thyrotropic hormone (TSH) concentrations were observed in the experimental group the AIN-93G (C) diet. The lowest TSH concentrations were observed in the experimental groups of rats that fed a diet with control green kale “Oldenbor F_1_” (CO) and biofortified 8-OH-7-I-5QSA kale (BO). Feeding of control (CR) and biofortified (BR) “Redbor F_1_” kale by Wistar rats resulted in slightly higher TSH hormone concentrations compared to “Oldenbor F_1_”, but not higher than AIN-93G (C). This indicates that iodine from non-biofortified and biofortified 8-OH-7-I-5QSA kale “Oldenbor F_1_” and “Redbor F_1_” cultivars was bioavailable to the rats and consequently was utilised for thyroid gland hormone synthesis. In contrast, elevated serum TSH levels in rats from the AIN-93G (C)—no kale supplementation groups, indicated insufficient iodine levels in this diet, compared to the rats’ requirement for this trace element. Thyroxine (T4) and triiodothyronine (T3) hormone concentrations were not significantly different between all experimental groups of rats tested. T3 is produced by deiodination of T4. The enzyme hepatic deiodinase type 1 is involved in this metabolic pathway. Furthermore, hepatic deiodinase type 1 is responsible for the serum T3 content [63,64,65]. It can be suggested that serum T4 concentrations in all groups of rats were not increased because T4 was used to produce T3.

The feed of different types of diets by Wistar rats did affect the change (*p* > 0.05) in the activity of the liver enzyme aspartate aminotransferase (AST) after 8 weeks of experimentation (Table 4). After feeding BR kale, a decrease in AST enzyme activity, or a tendency for BO to decrease, was observed. The feed of the kale cultivar “Redbor F_1_”, reduces AST enzyme concentrations the most. In the liver enzyme alanine aminotransferase (ALT), fed a diet with kale non-biofortified (CR) and -biofortified with 8-OH-7-I-5QSA (BR) “Redbor F_1_”, caused a significant reduction or a tendency for a lower, in the case of use of “Oldenbor F_1_” kale enriched in 8-OH-7-I-5QSA (BO), activity of the enzyme. In a study by Kopeć et al. [48] using lettuce biofortified with KI, enzyme concentrations did not change. However, in the study by Piątkowska et al. [55], ALT enzyme concentrations increased in the experimental groups where rats were fed cooked and KI-biofortified cooked carrots. It may be suggested that inorganic forms of iodine may influence increased liver activity and thus increase ALT enzyme activity. In contrast, in our experiment, the organic form of iodine in kale appears to be safer, as it caused a decrease in ALT activity. In addition, diets with red kale reduced ALT enzyme activity; this may be related to the high contents of bioactive compounds with liver-protective effects.

Total and direct bilirubin values were the highest in the group of rats fed the AIN-93G control diet (C). The other experimental groups fed non-biofortified (CO) and biofortified (BO, BR) kale had significantly lower—or a tendency for lower, in the case of CR—total and direct bilirubin values. Higher total and direct bilirubin values in group C may be indicative of liver problems (e.g., hepatitis, cirrhosis), blood diseases leading to increased breakdown of red blood cells (haemolysis), or bile duct problems (e.g., cholelithiasis). The addition of kale to the other CO, CR, BO, and BR diets showed a health-promoting effect through the polyphenols, glucosinolates, and dietary fibre present, which had a protective effect on liver function and thus reduced bilirubin absorption in the gastrointestinal tract [38].

The serum uric acid values of Wistar rats in our experiment were not affected by the diets. However, there was a trend towards lower uric acid concentrations in Wistar rats after feeding (non-) biofortified-with-8-OH-7-I-5QSA kale in both cultivars. Moreover, the lowest uric acid concentrations were in the groups where iodoquinoline was added. High levels of uric acid cause hyperuricaemia. Its higher levels sustained over a long period of time can result in the formation of sodium urate crystals and the development of diseases such as gout, kidney stones or cardiovascular conditions including hypertension [66]. Iodoquinol, which is a component of the tested compound 8-OH-7-I-5QSA, may contribute to the reduction of serum uric acid concentration in Wistar rats through various mechanisms. Although iodoquinol is mainly used as an antiparasitic agent in the treatment of amoebiasis [67], it may indirectly affect uric acid metabolism. Iodoquinol, by eliminating parasitic infections or reducing the microbial load, may reduce inflammation in the body. Chronic inflammation may interfere with purine metabolism, increasing uric acid production. By reducing inflammation, iodoquinol may normalize the breakdown of purines, leading to reduced serum uric acid production. Iodoquinol treatment (in the case of amoebiasis and other proven diseases [67] may improve general health or reduce oxidative stress in the kidneys, which may improve the glomerular filtration rate (GFR). Higher GFR means better removal of uric acid from the blood into the urine, which consequently lowers its concentration in the serum. Iodoquinol as an antiparasitic and antibacterial agent may alter the gut microbiome. The gut microbiota plays a role in nitrogen metabolism and purine processing. Changes in the microbial population may lead to altered purine metabolism, resulting in reduced uric acid production and absorption into the blood.

Changes in the HDL fraction cholesterol were observed. The highest HDL value was observed in rats fed AIN-93G (C) and in the CO, BO and BR groups. Furthermore, it is noteworthy that feeding of CO, BO, BR diets showed a tendency for a decrease in HDL concentrations, relative to the AIN-93G control. Significantly, the lowest HDL cholesterol concentration value was obtained by the research group feeding non-biofortified “Redbor F_1_” kale. This could be explained by the presence of fibre and other biologically active components. However, total cholesterol (TC) and LDL + VLDL were not affected by the different dietary treatments. Triglyceride concentrations had only a tendency for the highest concentration in the group of rats fed AIN-93G (C), compared to the other experimental groups (CO, BO, CR, BR). This may be explained by the presence of bioactive compounds in kale, which improve fat metabolism and thus faster excretion of fats with the gastrointestinal tract [68].

Antioxidant activity was determined in the serum of test rats by glutathione reductase (RG), total antioxidant status (TAS), and thiobarbituric acid reaction lipid peroxidation products (TBARs)—Table 4. After 8 weeks of the experiment, glutathione reductase levels were not significantly different between the experimental groups.

Determination of TAS in rat serum showed that feeding of “Oldenbor F_1_” kale (BO) and “Redbor F_1_” kale (BR) biofortified with 8-OH-7-I-5QSA significantly increased total antioxidant status. The increase in total antioxidant status may be due to the presence of numerous bioactive substances naturally present in kale. Furthermore, we can observe that the use of an iodoquinoline additive in the feed of rats also increased TAS concentrations. Iodoquinolines are a small number of quinoline derivatives that have been attributed with beneficial medical effects, including antibacterial, antifungal, antiviral, anti-inflammatory and anticancer effects [69]. The presence of these compounds may have a positive effect on increasing serum TAS concentrations in rats, but this requires further research.

Thiobarbituric acid reactive substances (TBARs) concentrations expressed in nmol malondialdehyde (MDA)·mL^−1^ were the highest in the AIN-93G control group (C). Research groups of rats which were fed diets supplemented with red kale of the “Redbor F_1_” cultivar (CR, BR) yielded the lowest values of serum MDA concentrations. Moreover, adding kale of the “Oldenbor F_1_” cultivar to the feed (CO, BO) caused a tendency for a reduction in MDA concentrations. MDA, which is an end product of lipid peroxidation, is commonly used as a marker of oxidative stress [70]. Kale has naturally occurring polyphenols with antioxidant activity. Polyphenols have been shown to have the ability to inhibit the increase in lipid peroxidation levels in laboratory animals [71]. It can be assumed that the effect of reducing serum MDA levels in rats fed non- and biofortified kale was due to the presence of polyphenols and iodoquinolines. Kopeć et al. [48] obtained similar results to ours. The addition of iodine-fortified lettuce (KI) and control lettuce significantly reduced serum TBARs in rats. Different results were presented by Piątkowska et al. [55], where the addition of control and cooked KI-biofortified carrots increased MDA concentrations. In the study by Rakoczy et al. [56], the use of KI-biofortified lettuce did not affect serum and liver MDA levels in rats.

The iodoquinoline biofortified kale cultivars, “Oldenbor F_1_” and “Redbor F_1_” can be considered as potential safe sources of iodine in the daily diet, preventing deficiencies of this trace element in different populations. In summary, the stages of the study and the final results are presented in Figure 3.

## 5. Conclusions

In conclusion, rats fed diets containing kale *Brassica oleracea* var. *sabellica* L. biofortified with iodine had higher iodine content in urine and kidneys, this proves the effective availability of iodine from kale biofortified with iodoquinolines, despite the presence of substances that hinder the absorption of iodine (goitrogens).

In rats feeding diets supplemented with “Oldenbor F_1_” and “Redbor F_1_” kale non- and -biofortified with 8-OH-7-I-5QSA, there were numerous health benefits through a tendency for a lower concentration of TSH, triglycerides, TBARs, total bilirubin, direct bilirubin, uric acid, aspartate aminotransferase (AST) and alanine aminotransferase (ALT) in the Wistar rats. Moreover, dietary intake of “Oldenbor F_1_” and “Redbor F_1_” kale biofortified with 8-OH-7-I-5QSA significantly increased total antioxidant status (TAS).

Biofortified kale *Brassica oleracea* var. *sabellica* L. can be considered as a potential source of iodine in the daily diet. It is reasonable to conduct more in vivo studies on this topic and to take the initiative to study the effects of consuming biofortified vegetables, including kale, with iodoquinolines on humans. Animal models, however, do not reflect bioavailability as reliably as studies involving humans. The results of biochemical tests and analyses of hormones in rat blood serum are not a reliable reflection compared to the analysis of human blood. On the other hand, the digestive tract of the Wistar rats is very similar to that of humans, so the bioavailability results may be highly reliable in planned human studies. The use of iodine-enriched kale diets has demonstrated effective bioavailability and absorption in the gastrointestinal tract of experimental rats. With the increasing popularity of kale as a superfood in recent years, there is a potential to enrich the daily diet with iodine by biofortified vegetables such as kale *Brassica oleracea* var. *sabellica* L. enriched with 8-OH-7-I-5QSA. The results of this research may lead to new health approaches that focus on increasing iodine availability, especially in regions with low levels of this element in the environment. Such initiatives may not only improve overall health but also reduce the risk of iodine-deficiency-related diseases, such as thyroid gland disorders, hypertension, obesity, and type 2 diabetes.

## 6. Patents

The method of biofortification of vegetables in iodine cultivated using traditional, soilless and hydroponic method and the use of 8-hydroxy-7-iodo-5-quinolinesulfonic acid for biofortification of vegetables with iodine. These are patent application number P.443218 (Polish Patent Office; 21 December 2022).

## Figures and Tables

**Figure 1 nutrients-16-03578-f001:**
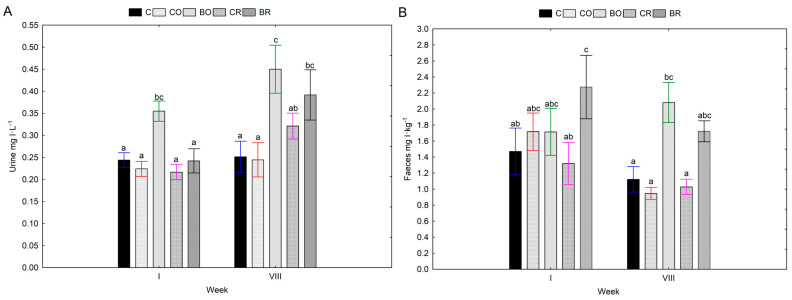
Concentrations of iodine in urine and faeces of Wistar rats. Contents of I in urine (**A**), contents of I in faeces (**B**); C—control diet (AIN-93G), CO—diet containing control curly kale “Oldenbor F_1_”, BO—diet containing biofortified curly kale “Oldenbor F_1_”, CR—diet containing control curly kale “Redbor F_1_”, BR—diet containing biofortified curly kale “Redbor F_1_”. Means followed by the same letters are not significantly different at *p* > 0.05 (Duncan’s post hoc test); bars indicate standard error (n = 8). Homogeneous groups refer to two-factor analysis of variance: Factor No. 1 type of diet: C, CO, BO, CR, BR; x Factor No. 2 week: I, VIII. The colour of the deviation line indicates the respective study group C (blue), CO (red), BO (green), CR (pink), BR (black).

**Figure 2 nutrients-16-03578-f002:**
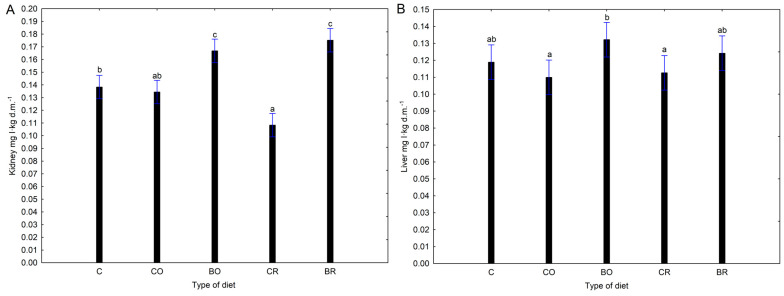
Concentration of iodine in selected organs. Contents of I in the kidney (**A**), contents of I in the liver (**B**); values in rows with different letters (a, b, c) are significantly different, *p* ≤ 0.05 (Duncan’s post hoc test); bars indicate standard error (n = 8). C—control diet (AIN-93G), CO—diet containing control curly kale “Oldenbor F_1_”, BO—diet containing biofortified curly kale “Oldenbor F_1_”, CR—diet containing control curly kale “Redbor F_1_”, BR—diet containing biofortified curly kale “Redbor F_1_”. Homogeneous groups refer to one-factor analysis of variance: Factor No.1 type of diet: C, CO, BO, CR, BR. The marking of the blue deviation line is the same for all experimental groups C, CO, BO, CR, BR.

**Figure 3 nutrients-16-03578-f003:**
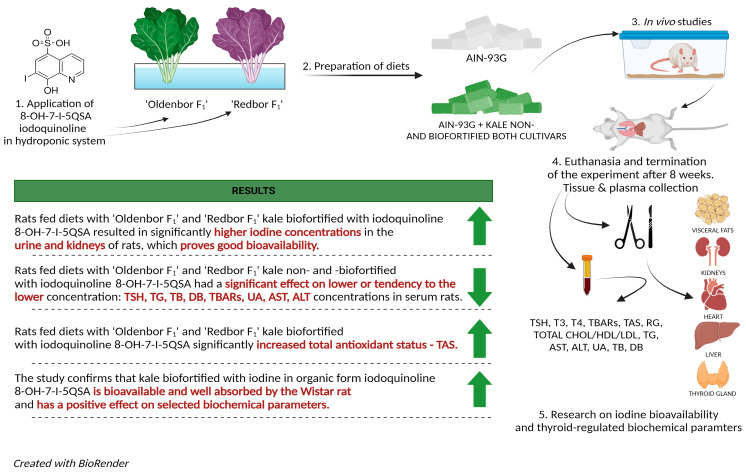
Graphic of study milestones and results.

**Table 1 nutrients-16-03578-t001:** Ingredients of experimental diets composition.

Ingredient (g·kg^−1^)	C	CO	BO	CR	BR
Corn starch	532.49	524.72	525.00	526.04	526.62
Saccharose	100	100	100	100	100
Casein	200	200	200	200	200
Soybean oil	70	70	70	70	70
Fibre	50	47.02	47.19	46.47	47
Vitamin mix ^a^	10	10	10	10	10
Mineral mix ^a^	35	35	35 ^b^	35	35 ^b^
Choline	2.5	2.5	2.5	2.5	2.5
TBHQ ^c^	0.014	0.014	0.014	0.014	0.014
Biofortified kale ^d^	-	-	10.3	-	8.87
Control kale ^d^	-	10.75	-	9.98	-

C—control diet (AIN-93G), CO—diet containing control curly kale “Oldenbor F_1_”, BO—diet containing biofortified curly kale “Oldenbor F_1_”, CR—diet containing control curly kale “Redbor F_1_”, BR—diet containing biofortified curly kale “Redbor F_1_”. ^a^ according to AIN-93G. ^b^ mineral mix without iodine; in these diets the source of iodine was biofortified curly kale in amounts providing an equivalent amount of iodine to the amount contained in AIN-93G. ^c^ tert-butylhydroquinone. ^d^ freeze-dried curly kale “Oldenbor F_1_” or “Redbor F_1_”.

**Table 2 nutrients-16-03578-t002:** Iodine content and basic chemical composition of kale both cultivars “Oldenbor F_1_” and “Redbor F_1_” used for preparation of experimental diets (g·100 g d.m.^−1^).

	Control Raw Kale “Oldenbor F_1_”	Raw Biofortified with 8-OH-7-I-5QSA Kale “Oldenbor F_1_”	Control Raw Kale “Redbor F_1_”	Raw Biofortified with 8-OH-7-I-5QSA Kale “Redbor F_1_”
Iodine (mg·kg d.m.^−1^)	0.18 ± 0.02 a	2.10 ± 0.04 b	0.20 ± 0.01 a	2.43 ± 0.08 c
Basic chemical composition				
Protein	32.98 ± 0.06 b	30.74 ± 1.06 b	26.03 ± 2.31 a	25.71 ± 0.88 a
Crude fat	5.46 ± 0.13 b	5.46 ± 0.26 b	3.24 ± 0.13 a	3.16 ± 0.15 a
Digestible carbohydrates	17.39 ± 0.77 a	20.39 ± 0.16 c	15.53 ± 0.43 b	18.35 ± 0.86 a
Dietary fibre	29.06 ± 0.82 a	28.45 ± 0.11 a	36.39 ± 0.33 c	34.71 ± 0.81 b
Ash	20.06 ± 0.23 b	19.21 ± 0.19 a	21.33 ± 0.09 d	20.76 ± 0.03 c

Values in rows with different letters (a, b, c, d) are significantly different, *p* ≤ 0.05 (Duncan’s post hoc test); bars indicate standard error (n = 4).

**Table 3 nutrients-16-03578-t003:** Body gain, fed efficiency ratio (FER), and weight of selected organs in fresh mass.

Type of Diet	C	CO	BO	CR	BR
Body gain (g)	302.38 ± 27.43 b	280.88 ± 26.62 ab	277.50 ± 17.65 a	273.38 ± 15.32 a	293.25 ± 20.10 ab
FER	0.202 ± 0.02 b	0.187 ± 0.02 ab	0.185 ± 0.01 a	0.182 ± 0.01 a	0.195 ± 0.01 ab
Liver (g)	16.14 ± 2.17 c	14.14 ± 1.52 b	13.04 ± 1.35 ab	12.32 ± 0.98 a	13.54 ± 1.80 ab
Kidney ** (g)	2.68 ± 0.14 a	2.47 ± 0.14 ab	2.55 ± 0.24 ab	2.29 ± 0.48 b	2.65 ± 0.24 a
Heart (g)	1.27 ± 0.09 b	1.18 ± 0.08 ab	1.22 ± 0.08 ab	1.14 ± 0.07 a	1.22 ± 0.08 ab
Thyroid gland (g)	0.23 ± 0.06 a	0.26 ± 0.03 ab	0.24 ± 0.04 ab	0.27 ± 0.03 ab	0.28 ± 0.03 b
Visceral fat (g)	4.39 ± 0.80 a	3.88 ± 0.68 a	4.18 ± 0.77 a	3.52 ± 1.06 a	3.77 ± 0.58 a

Values in rows with different letters (a, b, c) are significantly different, *p* ≤ 0.05 (Duncan’s post hoc test); bars indicate standard error (n = 8). ** weight of both kidneys. C—control diet (AIN-93G), CO—diet containing control curly kale “Oldenbor F_1_”, BO—diet containing biofortified curly kale “‘Oldenbor F_1_”, CR—diet containing control curly kale “Redbor F_1_”, BR—diet containing biofortified curly kale “Redbor F_1_”.

**Table 4 nutrients-16-03578-t004:** Selected biochemical parameters in serum of experimental rats.

Type of Diet	C	CO	BO	CR	BR
Liver panel					
AST (U·L^−1^)	36.11 ± 9.11 a	33.53 ± 12.39 a	30.54 ± 10.49 ab	31.11 ± 5.03 ab	19.79 ± 8.35 b
ALT (U·L^−1^)	26.55 ± 7.84 a	25.09 ± 6.81 a	23.74 ± 6.72 a	15.85 ± 4.84 b	13.93 ± 4.8 b
Bilirubin					
Total bilirubin (µmol·L^−1^)	10.38 ± 6.08 b	3.73 ± 1.84 a	3.15 ± 3.17 a	5.34 ± 3.47 a	4.28 ± 2.79 a
Direct bilirubin (µmol·L^−1^)	7.59 ± 3.85 b	3.35 ± 2.40 a	2.13 ± 1.27 a	4.53 ± 4.56 ab	4.27 ± 1.87 a
Uric acid					
Uric acid (µmol·L^−1^)	166.04 ± 62.01 a	132.35 ± 69.70 a	116.11 ± 67.30 a	112.50 ± 48.74 a	108.29 ± 19.76 a
Lipid profile					
TC (mmol·L^−1^)	3.01 ± 0.53 a	2.94 ± 0.22 a	2.77 ± 0.30 a	2.64 ± 0.34 a	2.55 ± 0.65 a
HDL (mmol·L^−1^)	1.89 ± 0.18 b	1.71 ± 0.22 ab	1.76 ± 0.34 ab	1.54 ± 0.27 a	1.64 ± 0.17 ab
LDL + VLDL (mmol·L^−1^)	1.12 ± 0.37 a	1.23 ± 0.31 a	1.01 ± 0.42 a	1.10 ± 0.20 a	0.91 ± 0.55 a
TG (mmol·L^−1^)	1.45 ± 0.62 b	1.35 ± 0.36 ab	0.99 ± 0.16 a	1.15 ± 0.44 ab	1.05 ± 0.22 ab
Antioxidant activity					
Glutathione reductase (U·L^−1^)	375.72 ± 205.75 a	348.31 ± 174.47 a	444.23 ± 83.52 a	378.96 ± 148.61 a	438.01 ± 68.74 a
TAS (mmol·L^−1^)	0.89 ± 0.20 ab	0.87 ± 0.08 a	1.04 ± 0.08 c	0.99 ± 0.10 abc	1.02 ± 0.10 bc
TBARs (nmol MDA·mL^−1^)	554.52 ±27.36 c	535.31 ± 88.82 bc	513.88 ± 49.46 abc	474.93 ± 63.19 ab	459.47 ± 66.32 a
Hormones					
TSH (ng·mL^−1^)	2.18 ± 0.21 b	2.02 ± 0.13 a	1.99 ± 0.08 a	2.10 ± 0.17 ab	2.07 ± 0.11 ab
T3 (pg·mL^−1^)	3.81 ± 0.19 ab	3.92 ± 0.15 b	3.73 ± 0.06 a	3.68 ± 0.09 a	3.84 ± 0.07 ab
T4 (ng·mL^−1^)	2.75 ± 0.59 a	2.87 ± 0.44 a	2.51 ± 0.53 a	2.59 ± 1.10 a	2.24 ± 0.84 a

Values in rows with different letters (a, b, c) are significantly different, *p* ≤ 0.05 (Duncan’s post hoc test); bars indicate standard error (n = 8). C—control diet (AIN-93G), CO—diet containing control curly kale “Oldenbor F_1_”, BO—diet containing biofortified curly kale “Oldenbor F_1_”, CR—diet containing control curly kale “Redbor F_1_”, BR—diet containing biofortified.

## Data Availability

The original contributions presented in the study are included in the article, further inquiries can be directed to the corresponding authors.

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
