# Peer review of "Iodine Bioavailability and Biochemical Effects of Brassica oleracea var. sabellica L. Biofortified with 8-Hydroxy-7-iodo-5-quinolinesulfonic Acid in Wistar Rats"

_nutrients, 2024, doi:10.3390/nu16213578_

Round 1
Reviewer 1 Report
Comments and Suggestions for Authors
This manuscript discusses the biofortification of kale with iodine and 8-hydroxy-7-iodo-5-quinolinesulfonic acid and its contribution to changes in thyroid-regulated parameters studied through animal studies. The topic of the work is suitable for publication in the journal. However, it is required that the authors include additional information and discussions in some sections. Several sentences are grammatically incorrect and should be rewritten or restructured.
I have the following comments to the authors:
1. Several sentences are wrongly structured, do not read well and should be rewritten:
· In lines 15-16, referring to the sentence: “biofortified with 8-hydroxy-7-iodo-5-quinolinesulfonic acid (8-OH-7-I-5QSA) kale”
This part of the sentence does not read well and should be restructured.
· In line 46, “,method is the of biofortified foods” is grammatically inaccurate and should be rewritten.
· In line 71, “contribute to increased of kale” is grammatically inaccurate and should be rewritten.
· In line 114, “will be used”, the verb tense should be corrected.
· In line 132, “males rats” is grammatically inaccurate and should be corrected.
· Lines 455-456, do not read well and should be rewritten.
· Lines 655-656 do not read well and should be rewritten.
2. In line 47, referring to “biofortification as a strategy” a definition of “biofortification” should be added here.
3. In section 2.1, provide an adequate reference for the protocol followed for hydroponic cultivation of kale.
4. In line 116, provide the complete term for: “TMAH”
5. Referring to line 163, add additional information on how rodents were killed. Was it according to ethical standards?
6. Relevant to lines 241-246, please add additional information to the main text on the reason for observing different ash or fiber content in different tested cultivars.
7. Referring to lines 268-269, add additional explanation to the main text on why kidney weight was lower for rats fed with non-biofortified kale.
8. Referring to lines 131-132, add a brief explanation on how iodoquinoline contributed to a decrease in uric acid concentration.
9. Referring to lines 394-396, please add a brief explanation on why higher iodine was accumulated in Redbor F1.
Comments on the Quality of English LanguageSeveral sentences are grammatically incorrect and wrongly structured. These sentences should be corrected or rewritten (see my enclosed comments).
Author Response
Dear Reviewer 1.
The entire team of co-authors would like to thank you very much for taking the time to do a thorough review of our manuscript. Each of the comments was extremely valuable to us
This manuscript discusses the biofortification of kale with iodine and 8-hydroxy-7-iodo-5-quinolinesulfonic acid and its contribution to changes in thyroid-regulated parameters studied through animal studies. The topic of the work is suitable for publication in the journal. However, it is required that the authors include additional information and discussions in some sections. Several sentences are grammatically incorrect and should be rewritten or restructured.
I have the following comments to the authors:
- Several sentences are wrongly structured, do not read well and should be rewritten:
- In lines 15-16, referring to the sentence: “biofortified with 8-hydroxy-7-iodo-5-quinolinesulfonic acid (8-OH-7-I-5QSA) kale”
This part of the sentence does not read well and should be restructured.
Response to the reviewer:
Thank you for your suggestion. The expression has been corrected.
- In line 46, “,method is the of biofortified foods” is grammatically inaccurate and should be rewritten.
Response to the reviewer:
Thank you for your suggestion. The expression has been corrected.
- In line 71, “contribute to increased of kale” is grammatically inaccurate and should be rewritten.
Response to the reviewer:
Thank you for your suggestion. The expression has been corrected.
- In line 114, “will be used”, the verb tense should be corrected.
Response to the reviewer:
Thank you for your suggestion. This subsection has been removed, in line with the comments of another reviewer.
- In line 132, “males rats” is grammatically inaccurate and should be corrected.
Response to the reviewer:
Thank you for your suggestion. The expression has been corrected.
- Lines 455-456, do not read well and should be rewritten.
Response to the reviewer:
Thank you for your suggestion. The expression has been corrected.
- Lines 655-656 do not read well and should be rewritten.
Response to the reviewer:
Thank you for your suggestion. The expression has been corrected.
- In line 47, referring to “biofortification as a strategy” a definition of “biofortification” should be added here.
Response to the reviewer:
Thank you for your suggestion. A definition of biofortification was added.
- In section 2.1, provide an adequate reference for the protocol followed for hydroponic cultivation of kale.
Response to the reviewer:
Thank you for your suggestion. This subsection has been removed, in line with the comments of another reviewer. A new text has been written with appropriate references.
- In line 116, provide the complete term for: “TMAH”
Response to the reviewer:
Thank you for your suggestion. This subsection has been removed in line with the comments of another reviewer, so the word ‘TMAH’ has been removed.
- Referring to line 163, add additional information on how rodents were killed. Was it according to ethical standards?
Response to the reviewer:
Thank you for your suggestion. Added information:
‘’Humane endpoints used in the study were administration of an analgesic (Butorphanol) and euthanasia by anesthetic overdose (isoflurane). Monitored symptoms: euthanasia of rats. After euthanasia, the animals will be transferred to the Sanitary Plants in Krakow, LLC, ul. Dymarek 7, 31-983 Krakow. Blood was collected by cardiac puncture into standard tubes.’’
- Relevant to lines 241-246, please add additional information to the main text on the reason for observing different ash or fiber content in different tested cultivars.
Response to the reviewer:
Thank you for your suggestion.
,,Ash content is the amount of minerals remaining after the organic matter has been burnt. The Redbor cultivar may have more efficient mechanisms for the uptake of nutrients such as calcium, potassium, magnesium or iron, leading to a higher ash content after burning. Redbor's dark purple colour is due to its higher content of anthocyanins (Zhang et al., 2012), which are mineral-rich compounds. Anthocyanins can promote a higher accumulation of certain minerals, which ultimately increases the ash content. Redbor, compared to Oldenbor, tends to have thicker and fleshy leaves, which may contain more nutrients and minerals, increasing the overall ash content. Redbor may be more efficient at taking up minerals from the soil compared to Oldenbor, which may be due to differences in genes, root system structure or soil preference. Redbor cultivar may have a thicker waxy coating on the leaves, which may contain minerals and contribute to higher ash content after burning. This coating has a protective function and, at the same time, may accumulate more minerals.’’
,,Kale cultivar Redbor is characterised by more robust, often wavy and thicker leaves. This leaf structure may be due to the higher content of cellulose and hemicellulose, which are the main fibre components. The genes responsible for cell wall thickness in Redbor may be more active than in Oldenbor, leading to higher fibre production. Redbor kale has a dark purple colour, indicating the presence of anthocyanins (Zhang et al., 2012), which can affect the structure and hardness of the leaves. An increase in lignin, which is also part of the fibre, may occur to strengthen the cell walls and protect the plant, increasing the total amount of fibre. Redbor kale often has crimped leaves, which may contain more structural tissue. This crimped texture suggests more structural fibre compared to the Oldenbor cultivar, which has a smoother texture. The Redbor cultivar is often more tolerant of harsh environmental conditions such as cold or drought, which may lead to increased production of lignin (a fibre component) in response to these stresses.’’
Zhang, B.; Hu, Z.; Zhang, Y. et al. A putative functional MYB transcription factor induced by low temperature regulates anthocyanin biosynthesis in purple kale (Brassica Oleracea var. acephala f. tricolor). Plant Cell Rep, 2012, 31, 281–289. https://doi.org/10.1007/s00299-011-1162-3
- Referring to lines 268-269, add additional explanation to the main text on why kidney weight was lower for rats fed with non-biofortified kale.
Response to the reviewer:
Thank you for your suggestion.
,,Lower kidney weights in Wistar rats fed non-biofortified kale (without iodoquinoline) may be due to several physiological and biochemical factors. Non-biofortified kale may contain fewer essential minerals and vitamins (such as iron, zinc and iodine), which are important for normal kidney function and growth. Biofortification usually increases the nutritional value of plants, including minerals as confirmed by the Krawczyk et al. (2019) kale study. The higher mineral content of biofortified kale may influence the greater growth of this organ. Non-biofortified kale may contain fewer antioxidants or other bioactive compounds that help alleviate oxidative stress. Without these compounds, the kidneys of rats consuming non-biofortified kale may be more susceptible to damage and cellular stress, leading to less organ weight. Kale that is non-biofortified may have a reduced ability to detoxify harmful substances. If a diet with non-biofortified kale has a lower content of detoxifying compounds, this may increase the metabolic load on the kidneys, causing renal stress and potential atrophy, which contributes to reduced kidney mass.’’
Krawczyk, K.; Smoleń, S.; Wisła-Świder, A.; Kowalska, I.; Kiełbasa, D.; Pitala, J.; Krzemińska, J.; Waśniowska, J.; Koronowicz, A. Kale (Brassica oleracea L. var. sabellica) biofortified with iodoquinolines: Effectiveness of enriching with iodine and influence on chemical composition. Scientia Horticulturae, 2024, 323, 112519. DOI: https://doi.org/10.1016/j.scienta.2023.112519
- Referring to lines 131-132, add a brief explanation on how iodoquinoline contributed to a decrease in uric acid concentration.
Response to the reviewer:
Thank you for your suggestion.
‘’Iodoquinol, which is a component of the tested compound 8-OH-7-I-5QSA, may contribute to the reduction of serum uric acid concentration in Wistar rats through various mechanisms. Although iodoquinol is mainly used as an antiparasitic agent in the treatment of amoebiasis (Mital, 2018), it may indirectly affect uric acid metabolism. Iodoquinol, by eliminating parasitic infections or reducing the microbial load, may reduce inflammation in the body. Chronic inflammation may interfere with purine metabolism, increasing uric acid production. By reducing inflammation, iodoquinol may normalize the breakdown of purines, leading to reduced serum uric acid production. Iodoquinol treatment (in the case of amoebiasis and other proven diseases (Mital, 2018)) may improve general health or reduce oxidative stress in the kidneys, which may improve the glomerular filtration rate (GFR). Higher GFR means better removal of uric acid from the blood into the urine, which consequently lowers its concentration in the serum. Iodoquinol as an antiparasitic and antibacterial agent may alter the gut microbiome. The gut microbiota plays a role in nitrogen metabolism and purine processing. Changes in the microbial population may lead to altered purine metabolism, resulting in reduced uric acid production and absorption into the blood.’’
Mital, A. (2018). Amoebiasis Revisited. In: Singh, P. (eds) Infectious Diseases and Your Health. Springer, Singapore. https://doi.org/10.1007/978-981-13-1577-0_2
- Referring to lines 394-396, please add a brief explanation on why higher iodine was accumulated in Redbor F1.
Response to the reviewer:
Thank you for your suggestion.
‘’The Redbor kale cultivar can accumulate more iodine than other cultivars, such as Oldenbor, due to several key factors related to genetics, leaf structure and the plant's ability to absorb minerals from the soil. Kale cultivars can vary in the structure and efficiency of their root system, which affects their ability to absorb minerals, including iodine, from the soil. The Redbor cultivar may have a more developed root system better adapted to efficiently take up iodine from the soil. The Redbor genotype may influence the higher activity of enzymes and proteins responsible for the transport and storage of iodine in plant cells. Each kale cultivar is genetically different and Redbor may be more predisposed to accumulate higher amounts of iodine. This requires further research. The Redbor cultivar is rich in anthocyanins (phenolic compounds) (Zhang et al., 2012) which may affect the absorption and transport of minerals in the plant. These compounds can promote the storage of trace elements, including iodine, leading to greater iodine accumulation compared to varieties with lower anthocyanin content, such as Oldenbor. Anthocyanins can protect the plant against oxidative stress and UV radiation (Zhang et al., 2012), which can improve the efficiency of mineral uptake, including iodine, from the soil. Redbor, with its more developed, thicker leaves, can better manage transpiration, which promotes more efficient transport of iodine to the leaves.’’
Zhang, B.; Hu, Z.; Zhang, Y. et al. A putative functional MYB transcription factor induced by low temperature regulates anthocyanin biosynthesis in purple kale (Brassica Oleracea var. acephala f. tricolor). Plant Cell Rep, 2012, 31, 281–289. https://doi.org/10.1007/s00299-011-1162-3
Comments on the Quality of English Language
Several sentences are grammatically incorrect and wrongly structured. These sentences should be corrected or rewritten (see my enclosed comments).
Response to the reviewer:
Thank you for your suggestion. The article was submitted for linguistic revision. A certificate of linguistic correctness of the manuscript is attached.

Reviewer 2 Report
Comments and Suggestions for Authors
The current manuscript is on a continuum of a long-term programme of research on Brassica oleracea L. var. sabellica biofortified with iodoquinolines (https://doi.org/10.1016/j.scienta.2023.112519; https://doi.org/10.3390/nu15224730 and https://doi.org/10.1371/journal.pone.0304005). As it is, the manuscript needs to be revised substantially to improve its quality and align it with scientific guidelines of reporting. I feel that the draft suffers significantly from self-plagiarism or other such vice as duplicate, multiple or redundant publication. My specific comments are detailed in the following:
1. The present title is too long. It could be revised to precisely read ‘‘Iodine Bioavailability and Biochemical Effects of Brassica oleracea L. Biofortified with 8-hydroxy-7-iodo-5-quinolinesulfonic acid in Wistar Rats". I would encourage the authors to use the botanical name of the species rather than ‘‘kale’’ throughout the manuscript.
2. In the abstract (L15-L17), you need to give a brief introduction then the methods followed before giving results. It is important that you indicate which P-value the statistical significance is being considered. L24 should introduce some perspective(s) for further research.
3. The keywords suggested should not repeat words that are part of the title. This way, you will increase the visibility of the final published article.
4. The introduction needs to be improved.
(i) The problem is probably not iodine but its deficiency. In this case, you should briefly mention on symptoms of iodine deficiency and related thyroid disorders. The text should then conduct the reader to the various measures of countering iodine deficiency in the population. This is a joint global effort as far as I know, including the iodization of salt.
(ii) Since your previous publications cited above are relevant for understanding the history of the accumulated record, it is recommended that you introduce what you have done so far towards the end of the introduction.
5. The present submission has upto 37% overlap with previously published literature, especially the METHODS and RESULTS. This could constitute self-plagiarism or other such vice as duplicate, multiple or redundant publication. This is also why there is a need to delineate the present submission from previous research efforts as indicated in comment 4(ii). I would recommend citing those studies you already described some of the methods in than presenting all the methods here without a single reference.
6. In L106, data from experiments involving animals are best presented as mean±standard error of replicates because this provides a clear understanding of the average outcome (mean) and the variability or precision of that average (standard error). I see that this is indicated in L232, and somehow introduces contradiction in the study. Please recheck.
It is not immediately clear why Duncan's test was preferred in L210. This posthoc test is considered less conservative compared to others like Tukey's, which means it has a higher chance of detecting significant differences but also a higher risk of Type I errors (false positives). You may wish to point out (if at all) some means were also statistically significant at the P<0.01 level.
7. The equation in L255 should be defined under METHODS.
8. OTHER SUGGESTIONS
-The RESULTS should be presented in a continuous but logical manner. In the current draft, they are presented as short paragraphs.
-L676-L677 is unnecessary.
-The pictorial flow of procedures provided in the supplementary materials could be placed in the main text to save the high similarity percentage due to reuse of previously published methods word-for-word.
-Some of the Tables could be visualized as graphs or charts.
Comments on the Quality of English LanguageMinor grammatical fixes required
Author Response
Dear Reviewer 2.
The entire team of co-authors would like to thank you very much for taking the time to do a thorough review of our manuscript. Each of the comments was extremely valuable to us. We have done our best to make corrections and adequately justify why certain situations occurred.
The current manuscript is on a continuum of a long-term programme of research on Brassica oleracea L. var. sabellica biofortified with iodoquinolines (https://doi.org/10.1016/j.scienta.2023.112519; https://doi.org/10.3390/nu15224730 and https://doi.org/10.1371/journal.pone.0304005). As it is, the manuscript needs to be revised substantially to improve its quality and align it with scientific guidelines of reporting. I feel that the draft suffers significantly from self-plagiarism or other such vice as duplicate, multiple or redundant publication. My specific comments are detailed in the following:
- The present title is too long. It could be revised to precisely read ‘‘Iodine Bioavailability and Biochemical Effects of Brassica oleracea L. Biofortified with 8-hydroxy-7-iodo-5-quinolinesulfonic acid in Wistar Rats". I would encourage the authors to use the botanical name of the species rather than ‘‘kale’’ throughout the manuscript.
Response to the reviewer:
- Thank you for your suggestion. The title has been changed as per the reviewer's suggestion. However, we will also add the var. sabellica Since we want to stick to one style with previously published articles and in accordance with the recommendations from:
https://powo.science.kew.org/taxon/urn:lsid:ipni.org:names:60452375-2
Brassica oleracea var. sabellica L.
- Thank you for your suggestion. Increased use of the botanical name instead of kale in text.
- In the abstract (L15-L17), you need to give a brief introduction then the methods followed before giving results. It is important that you indicate which P-value the statistical significance is being considered. L24 should introduce some perspective(s) for further research.
Response to the reviewer:
- Thank you for your suggestion, added L15-L17: ,,Iodine is one of the essential trace elements for human life. The main objective of the biofortification of plants with iodine, is to obtain food with a higher content of this element compared to conventional food. Biofortification of plants with iodine, can increase the intake of this trace element by different populations. In addition, it may reduce the risk of iodine deficiency diseases.”
- Thank you for your suggestion. Added P-value.
- Thank you for your suggestion, added L24: ,, Initiative to study the effect of human of kale biofortified with iodoquinolines should be undertaken. Animal models, however, do not reflect bioavailability as reliably as studies involving humans. The results of biochemical tests and analyses of hormones in rat blood serum are not a reliable reflection compared to the analysis of human blood. On the other hands, due to the fact that the digestive tract of the Wistar rat is very similar to that of humans, the bioavailability results may be highly reliable in planned human studies.”
- The keywords suggested should not repeat words that are part of the title. This way, you will increase the visibility of the final published article.
Response to the reviewer:
- Thank you for your suggestion. Removed repeated words in title. Added new keywords.
- The introduction needs to be improved.
(i) The problem is probably not iodine but its deficiency. In this case, you should briefly mention on symptoms of iodine deficiency and related thyroid disorders. The text should then conduct the reader to the various measures of countering iodine deficiency in the population. This is a joint global effort as far as I know, including the iodization of salt.
Response to the reviewer:
- Added about iodine deficiency and the consequences (L37-L45), and in further paragraphs (L55-L73) it is stated that biofortification of plants, iodisation of salt or general supplementation prevents deficiencies of this element.
(ii) Since your previous publications cited above are relevant for understanding the history of the accumulated record, it is recommended that you introduce what you have done so far towards the end of the introduction.
- Thank you for your suggestion. Added some information about earlier research.
- The present submission has upto 37% overlap with previously published literature, especially the METHODS and RESULTS. This could constitute self-plagiarism or other such vice as duplicate, multiple or redundant publication. This is also why there is a need to delineate the present submission from previous research efforts as indicated in comment 4(ii). I would recommend citing those studies you already described some of the methods in than presenting all the methods here without a single reference.
- Response to the reviewer:
Thank you for your suggestion. Subsection 2.1. and 2.2. has been removed and the methodology has been cited using an earlier publication.
Deleted L77 – L121.
- In L106, data from experiments involving animals are best presented as mean±standard error of replicates because this provides a clear understanding of the average outcome (mean) and the variability or precision of that average (standard error). I see that this is indicated in L232, and somehow introduces contradiction in the study. Please recheck.
- Response to the reviewer:
Thank you for your suggestion. Verified and corrected.
It is not immediately clear why Duncan's test was preferred in L210. This posthoc test is considered less conservative compared to others like Tukey's, which means it has a higher chance of detecting significant differences but also a higher risk of Type I errors (false positives). You may wish to point out (if at all) some means were also statistically significant at the P<0.01 level.
- Response to the reviewer:
Thank you for your suggestion. The reviewer's remark is correct, however, we always perform a preliminary analysis with various tests i.e. Duncan, Tukey, and Fisher's for a thorough comparison before doing the final presentation of statistical results. In this case, our analysis showed that there were no significant differences between Duncan's test and Tukey's test. Therefore, Duncan's test was applied concerning previously published papers to maintain homogeneous continuity and style. We include a screenshot of the ANOVA results of the AST enzyme analysis about dietary intake using the two tests (Duncan’s and Tukey), where no differences were shown. To be sure, we present a few examples. In other analyses, the results were the same e.g. uric acid screen (in Polish: ,,kwas moczowy”).
- The equation in L255 should be defined under METHODS.
Response to the reviewer:
Removed in L255, added to MATERIAL AND METHODS chapter in L182.
- OTHER SUGGESTIONS
- The RESULTS should be presented in a continuous but logical manner. In the current draft, they are presented as short paragraphs.
Response to the reviewer:
Thank you for your suggestion. A few introductory sentences have been added in subsections and the text has been diversified.
- L676-L677 is unnecessary.
Response to the reviewer:
Thank you for your suggestion. We may remove this funding information if the reviewer is very concerned about it or if the information is shown elsewhere. If they are not shown elsewhere, the entire team prefers not to remove them because we want the funding sources to be open and clear.
- The pictorial flow of procedures provided in the supplementary materials could be placed in the main text to save the high similarity percentage due to reuse of previously published methods word-for-word.
Response to the reviewer:
Thank you for your suggestion. Added image in the discussion section to summarise and add variety to the text.
- Some of the Tables could be visualized as graphs or charts.
Response to the reviewer:
Thank you for your suggestion. Changed Table 4. to graph Figure 2.
- Comments on the Quality of English Language: Minor grammatical fixes required
Response to the reviewer:
Thank you for your suggestion. The article was submitted for linguistic revision. A certificate of linguistic correctness of the manuscript is attached.

Reviewer 3 Report
Comments and Suggestions for Authors
The proposed manuscript presents interesting data regarding the possibilities of fortifying plant sources used for food with iodine. In this way, various nutritional deficiencies could be corrected, including iodine deficiency and the risk of thyroid disease. I do not doubt the scientific value of the research and experiments conducted. But the Manuscript needs serious corrections and revision.
1. The English language must be edited, preferably by a native English speaker. In some places in the text, it is not clear what the authors want to say.
2. It is not clear from the experimental design whether the control group of animals is a pure control or an iodine deficiency control. If it is a disease control, a normal control, rats fed normal laboratory rodent chow, should also be included in the experiment. If this is a control group without any deficiency or other pathology, there should also be a group in the experiment representing a disease model, ie. positive control. Depending on this, the results and the discussion would be interpreted differently. Assuming this is a normal control, why are bilirubin, MDA, and transaminases elevated in this group? Increased activity of ALT and AST and increased levels of bilirubin are present in the presence of liver damage. An elevated level of MDA is a sign of marked oxidative stress and lipid peroxidation. What is the cause of these violations? The content of the various components in the control diet was not much different from the other diets. Why are these deviations observed in the biochemistry of animals?
Comments on the Quality of English LanguageThe English language must be edited, preferably by a native English speaker or professional translator. In some places in the text, it is not clear what the authors want to say. For example: "Rats diets with 'Oldenbor F1' and 'Redbor F1' kale non- and -biofortified with 8-OH-7- 18 I-5QSA had a significantly lower or a tendency for lower concentration of TSH, triglyceride........". MS is full of such examples.
Author Response
Dear Reviewer 3.
The entire team of co-authors would like to thank you very much for taking the time to do a thorough review of our manuscript. Each of the comments was extremely valuable to us. We have done our best to make corrections and adequately justify why certain situations occurred.
The proposed manuscript presents interesting data regarding the possibilities of fortifying plant sources used for food with iodine. In this way, various nutritional deficiencies could be corrected, including iodine deficiency and the risk of thyroid disease. I do not doubt the scientific value of the research and experiments conducted. But the Manuscript needs serious corrections and revision.
- The English language must be edited, preferably by a native English speaker. In some places in the text, it is not clear what the authors want to say.
Response to the reviewer:
The article was submitted for linguistic revision. A certificate of linguistic correctness of the manuscript is attached.
- It is not clear from the experimental design whether the control group of animals is a pure control or an iodine deficiency control. If it is a disease control, a normal control, rats fed normal laboratory rodent chow, should also be included in the experiment. If this is a control group without any deficiency or other pathology, there should also be a group in the experiment representing a disease model, ie. positive control. Depending on this, the results and the discussion would be interpreted differently. Assuming this is a normal control, why are bilirubin, MDA, and transaminases elevated in this group? Increased activity of ALT and AST and increased levels of bilirubin are present in the presence of liver damage. An elevated level of MDA is a sign of marked oxidative stress and lipid peroxidation. What is the cause of these violations? The content of the various components in the control diet was not much different from the other diets. Why are these deviations observed in the biochemistry of animals?
Response to the reviewer:
- The control group (C) was fed the AIN-93G diet, in which the reference amounts of iodine are derived from an inorganic compound (KIO3). The situation is similar in human nutrition, where the most common form of iodine source is KO3 or KI (inorganic forms) contained in fortified table salt, the intake of which should be limited for known reasons, consequently iodine intake would be insufficient. The present study did not focus on the effects of iodine deficiency, as these have been well studied and described in the world literature. Instead, the aim of the present study was to comparatively assess the effects on the body of iodine derived from the organic form (iodoquinoline) with which kale was biofortified, in relation to the inorganic form used (group C). As demonstrated earlier, this compound is efficiently incorporated into the plant and can be an alternative to iodised table salt. The results obtained are intended to answer the question of whether kale (and perhaps other biofortified vegetables) can be a substitute for the iodine carrier table salt.
- In the case of laboratory rats, there are no reference values ​​for biochemical and other parameters. The relevant parameters are those of the control group (C), to which the results of all other tested groups of rats are compared. In this case, the above-mentioned parameters (MDA, bilirubin, AST, ALT) in rats fed a diet with the addition of red kale (group CR and BR) were favorably reduced in relation to the control group C, which was fed the AIN-93G diet. The probable cause of the reduction in the parameter mentioned above values ​​was the addition of kale to the basic AIN-93G diet - a rich carrier of many bioactive compounds. A more extensive explanation is included in the discussion of the results. It would be interesting to trace the mechanisms of the observed changes, as a subject of further studies.
The explanation of the observed effects is included in the discussion.
Comments on the Quality of English Language
The English language must be edited, preferably by a native English speaker or professional translator. In some places in the text, it is not clear what the authors want to say. For example: "Rats diets with 'Oldenbor F1' and 'Redbor F1' kale non- and -biofortified with 8-OH-7- 18 I-5QSA had a significantly lower or a tendency for lower concentration of TSH, triglyceride........". MS is full of such examples
- Response to the reviewer:
The article was submitted for linguistic revision. A certificate of linguistic correctness of the manuscript is attached.

Round 2
Reviewer 2 Report
Comments and Suggestions for Authors
The authors have responded to all the comments I had on the previous draft. I now suggest removing the current supplementary files as it is now part of the main manuscript (Figure 3). The long title of the figure Given in UPPER CASE should be deleted since the figure now has a caption.
Comments on the Quality of English LanguageOnly minor corrections necessary to polish the draft
Author Response
Thank you for your valuable reviews.
Reviewer 3 Report
Comments and Suggestions for Authors
Dear Authors,
Thank you for your cooperation!
Author Response
Thank you for your valuable reviews.